# Analysis and Impact of Activated Carbon Incorporation into Urea-Formaldehyde Adhesive on the Properties of Particleboard

Mehmet Emin Ergun [1], İsmail Özlüsoylu [2,*], Abdullah İstek [2] and Ahmet Can [2,*]

1    Akseki Vocational High School, Alanya Alaaddin Keykubat University, Antalya 07630, Turkey; mehmet.ergun@alanya.edu.tr
2    Faculty of Forestry, Bartin University, Bartin 74100, Turkey; aistek@bartin.edu.tr
*    Correspondence: iozlusoylu@bartin.edu.tr (İ.Ö.); acan@bartin.edu.tr (A.C.)

**Abstract:** Nowadays, the particleboard industry cannot meet the market's demand. Therefore, filler materials have started to be used both to conserve raw materials and to enable the use of wood-based boards in different areas. This study investigates the effects of incorporating different ratios of activated carbon (0%, 1.5%, 4.5%, 7.5%) on the properties of particleboards. The physical properties were examined, including density, moisture content, thickness swelling, and water absorption. The results reveal that the density increased with increasing activated carbon content while the moisture content decreased, indicating improved dimensional stability and water resistance. Additionally, the color properties were influenced by activated carbon, leading to a darker appearance with decreased lightness and yellow-blue components. The mechanical properties, such as internal bond strength, modulus of rupture, and modulus of elasticity, showed significant enhancements with the addition of activated carbon, indicating improved bonding and increased strength. Moreover, the thermal conductivity decreased with increasing activated carbon content and improved insulation performance. Scanning electron microscope analysis confirmed the uniform distribution of activated carbon within the particleboard matrix, without agglomeration, positively impacting the mechanical performance. According to the thermogravimetric analysis results, the addition of activated carbon led to a decrease of up to 6.15% in mass loss compared to the control group. The incorporation of activated carbon at a ratio of 4.5% in particleboards confers notable enhancement to their physical, mechanical, and thermal characteristics. These findings contribute to understanding the potential benefits and considerations of using activated carbon as an additive in particleboard production.

**Keywords:** activated carbon; additive; particleboards; mechanical properties; physical properties



## 1. Introduction

Wood is a preferred building and engineering material due to its properties, such as being a good insulator, easy to work with, and aesthetically pleasing, and not causing any harm to humans or the environment during both its use and after its service life [1]. However, due to the increasing population and demands, solid wood alone cannot meet the required needs, making it necessary to utilize wood waste [2]. Adding adhesives transforms wood waste into products such as fiberboard and particleboard. Particleboard, born out of the need to utilize large quantities of wood waste, such as sawdust, planer shavings, and, to a lesser extent, mill residues generated by other wood industries, has become increasingly popular. It is widely used in manufacturing furniture, cabinets, and various structural products [3]. In 2021, 4,075,000 m$^3$ and 103,955,051 m$^3$ of particleboard were produced in Turkey and the world, respectively [4]. Due to the particleboard industry's inability to meet market demand, filler materials are being used to conserve raw materials and enable the use of wood-based boards in different areas [5]. Thus, filler materials, such as vermiculite [6] for improving acoustic properties, calcium carbonate [7], clay [8], and montmorillonite [9] for enhancing thermal properties, carbon nanotubes [10] for improving

physical properties, and zeolite [11] and activated carbon [12] for enhancing mechanical properties, have been used in previous studies.

Activated carbon is an amorphous material obtained by activating high-carbon content organic and inorganic materials through chemical or physical methods to increase their internal surface area and pore volume [13]. Furthermore, activated carbon differs from elemental carbon as it allows for widespread use in adsorption and purification due to the oxidation of carbon atoms present on its internal and external surfaces [14]. Additionally, activated carbon is used in capacitors [15], and recently, in composite materials [16]. It was found that materials produced from silicone-modified activated carbon exhibited resistance against combustion [17]. Activated carbon aerogels derived from carboxymethyl cellulose could be used in energy storage applications [18]. The addition of activated carbon to poly(lactic acid)-polybutylene succinate foams enhanced their thermal and mechanical properties [19]. In the last few decades, activated carbon has started to be studied in fiberboard production. Studies have examined the effects of adding activated carbon on the physical properties (thickness swelling, water absorption, and density) and mechanical properties (internal bond strength, modulus of rupture, and modulus of elasticity) of produced medium-density fiberboard [12,20]. The porous and diverse functional group structure of activated carbon was utilized to reduce the emission of free formaldehyde in wood-based panels [21]. Wood-based panels are primarily used as construction materials indoors. Therefore, investigating the thermal properties of these materials and elucidating the impact of activated carbon is crucial. In addition, the effects of activated carbon have not been investigated before on particleboard properties.

This study explored the effects of adding activated carbon at different ratios to single-layer particleboard on its physical, mechanical, thermal, and morphological properties. Thus, the usability of activated carbon as an additive in particleboard production was investigated. In addition, the thermal properties of particleboards, mostly used indoors, were examined, and the thermal effect of activated carbon was determined.

## 2. Materials and Methods

### 2.1. Materials

Activated carbon (AC) was bought from Aromel Chemie (Konya, Turkey) and possessed a 2140 $kg/m^3$ density. The AC was produced from coconut shells with the physical activation method. The commercial particleboard factory supplied urea-formaldehyde adhesive (UF) and the chips. The dimensions of the chips were 10.5 mm × 10.5 mm. The chip mixture ratios were as follows: *Pinus nigra*: 50%, *Fagus orientalis*: 20%, *Carpinus betulus*: 20%, *Populus nigra*: 10%. The UF was characterized by a viscosity of 220 mPa·s at 22 °C, a density ranging from 1260 $kg/m^3$ to 1280 $kg/m^3$, and a solid content of approximately 65%. To facilitate adhesive curing, a 20% ammonium chloride solution was added with a proportion of 1.5% relative to the adhesive weight, and the solution was stirred at 600 rpm for 10 min with a mechanical stirrer (WiseStir, HS-120A, Daihan Scientific, Gangwon, South Korea). All chemicals supplied were of analytical grade and procured from Merck (Darmstadt, Germany) or Fluka (Buchs, Switzerland).

### 2.2. Production of Particleboard

The chips obtained from the particleboard facility produced single-layer particleboards. Before adhesive application, the chips were dried to a moisture content ranging from 1% to 3% and stored in airtight plastic bags to prevent air exposure. The study employed UF with a solid content of 65%. AC was incorporated into the adhesive at concentrations of 1.5%, 4.5%, and 7.5% based on the dry matter content of the adhesive, and the mixture was stirred at 600 rpm for 30 min. The viscosities of the adhesive were 0 % AC: 242 ± 4, 1.5% AC: 255 ± 3, 4.5% AC: 267 ± 6, and 7.5% AC: 271 ± 5 at 22 °C. The adhesive application was conducted using an adhesive gun within a rotary drum mixer (ECOMIX, 42750 Saint-Denis de Cabanne). Subsequently, the manual spreading process was performed using a wooden spreading mold, and the board assembly was placed into a hot press with dimensions of

$400 \times 400 \times 12$ mm. Each group produced three particleboards. After the pressing process, the resulting boards were allowed to cool and be conditioned at a relative humidity of $65 \pm 5\%$ and a temperature of $20 \pm 2$ °C to achieve equilibrium moisture content. The samples were then extracted for the determination of board properties. The particleboard production parameters are shown in Table 1.

**Table 1.** Particleboard production parameters.

| Particleboard Production Parameters | Values |
|---|---|
| Thickness (mm) | 12 |
| Target board density (kg/m$^3$) | 700 |
| Board dimensions (mm) | $400 \times 400$ |
| Specific press pressure (bar) | 35 |
| Press temperature (°C) | 175 |
| Press time (s) | 220 |

*2.3. Characterization*

The particle size distribution of the AC was measured with a Mastersizer 3000 instrument (Malvern, Cambridge, United Kingdom). The test panels were left to stabilize under ambient conditions for one week. Following this, the test samples were prepared in accordance with the EN 326-1 (1999) standard [22]. The samples were then conditioned in a controlled environment chamber for one week at $20 \pm 2$ °C and a relative humidity of $65 \pm 5\%$. The moisture content (MC) of the test panels was determined according to the EN 322 (1999) standard [23], while their densities were measured according to EN 323 (1999) [24]. Thickness swelling (TS) was evaluated following the guidelines of EN 317 (1999) [25]. Water absorption (WA) was conducted according to ASTM D1037-12 (2020) [26]. The color measurements of the particleboards were carried out using a Konica Minolta spectrophotometer (Osaka, Japan). According to ISO 7724-2 (1984) [27], the L*, a*, and b* values were determined by analyzing the measurements taken from randomly chosen areas. The evaluation of the color coordinates used the CIE Lab system. The experimental boards' L*, a*, and b* color coordinates for each variation were established for particleboards both with and without AC. The L* axis represents the black–white axis, where L* = 0 corresponds to black and L* = 100 corresponds to white. The a* axis represents the red–green color spectrum, with positive values indicating red and negative values indicating green. The b* axis represents the yellow–blue color spectrum, with positive values indicating yellow and negative values indicating blue. Changes in the color coordinates (ΔL*, Δa*, and Δb*) were calculated by determining the difference between the values obtained by the control group (0% AC) and the particleboard, which included different ratios of AC. The total color differences (ΔE*) were calculated according to Equation (1).

$$\Delta E^* = [(\Delta a^*)^2 + (\Delta b^*)^2 + (\Delta L^*)^2]^{1/2} \tag{1}$$

where ΔL* represents the change in L*, Δa* represents the change in a*, and Δb* represents the change in b*. ΔE* represents the total color change.

The internal bond strength (IB), modulus of rupture (MOR), and modulus of elasticity (MOE) were determined according to EN 319 (1999) [28] and EN 310 (1999) [29], respectively. The morphologies of the particleboards were examined using a scanning electron microscope (SEM) (Maia3 Xmu model, TESCAN, Brno, Czech Republic). In order to enhance the conductivity of the particleboards, a thin layer of gold with a thickness of 5 nm was applied to the samples. The SEM microscope was operated at a voltage of 20.0 kV during the analysis of microstructure images. The thermal conductivity of the particleboards was measured at 24 °C, and the ASTM C518 (2021) standard [30] was followed, utilizing a heat flux instrument (KEM QTM 700, Kyoto Electronics, Kyoto, Japan). The samples were subjected to thermogravimetric (TG) and derivative thermogravimetric (DTG) analyses (Hitachi-STA 7300, Hitachi, Ltd., Tokyo, Japan), where they were heated

from room temperature to 600 °C. The heating rate used was 10 °C/min under a nitrogen atmosphere, with a 50 mL/min gas flow rate. All tests, except for the TGA and SEM analyses, were repeated in six samples for each group. The impact of varying amounts of activated carbon on the physical and mechanical properties of the chipboard was assessed using a one-way analysis of variance (ANOVA) in the SPSS 16 program, with a confidence level of 95% ($p < 0.05$). Statistically significant homogeneous groups were determined using the Duncan homogeneity test.

## 3. Results and Discussion

### 3.1. Particle Size Distribution of the Activated Carbon

The particle size distribution of the AC, which was essential to determine before utilizing it in particleboard production, is shown in Figure 1.

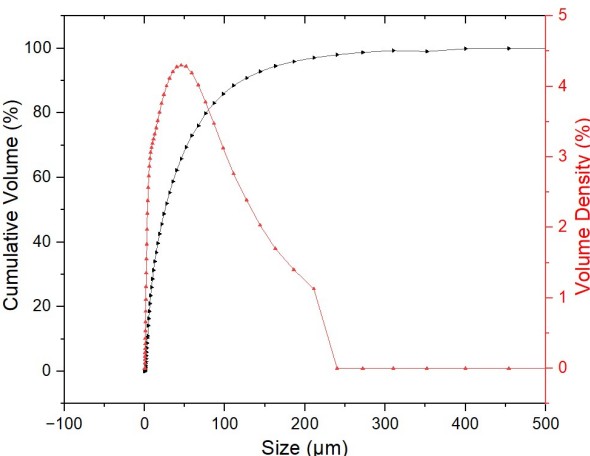

**Figure 1.** Particle size distribution of AC.

It was determined that 95% of the particle sizes of AC were smaller than 60 μm. The particle size was closely associated with grinding, which constituted the final stage of AC production. Fourier transform infrared spectroscopy, field emission scanning electron microscopy, energy dispersive spectrometry, and elemental mapping analyses of activated carbon were given in Ergun and Bulbul's study [31].

### 3.2. Physical Properties of Particleboards

The statistical analysis of the physical properties of the produced single-layered boards is shown in Table 2. The letters (a, b, and c) in Table 2 indicate that sample groups with equal importance are clustered.

**Table 2.** Physical test results of AC-reinforced particleboards.

| Codes | Density (kg/m³) | MC (%) | TS (%) (2 h) | TS (%) (24 h) | WA (%) (2 h) | WA (%) (24 h) |
|---|---|---|---|---|---|---|
| %0 AC | 750 [a] ± 53.87 | 7.13 [b] ± 0.41 | 26.18 [a] ± 1.92 | 37.42 [ab] ± 2.66 | 49.81 [a] ± 5.15 | 66.38 [a] ± 6.99 |
| %1.5 AC | 750 [a] ± 46.60 | 6.25 [a] ± 0.33 | 31.27 [b] ± 3.27 | 40.18 [bc] ± 2.41 | 49.49 [a] ± 4.01 | 66.45 [a] ± 3.82 |
| %4.5 AC | 760 [a] ± 47.03 | 6.24 [a] ± 0.07 | 26.53 [a] ± 1.84 | 35.11 [a] ± 4.37 | 49.31 [a] ± 5.01 | 65.34 [a] ± 4.01 |
| %7.5 AC | 770 [a] ± 44.28 | 6.17 [a] ± 0.01 | 29.43 [ab] ± 4.24 | 42.32 [c] ± 4.08 | 47.43 [a] ± 3.74 | 64.16 [a] ± 4.26 |

[a,b,c] Letters indicate Duncan's homogeneity groups in the column.

The results show that the AC content affected the physical properties of the particleboard. When the effects of AC addition on some physical properties of the particleboards were examined statistically, it was understood that there was a significant difference between the 2 h and 24 h TS values ($p < 0.05$). Although the AC content increased in the

particleboards, and the density of the particleboards changed between 750 kg/m$^3$ and 770 kg/m$^3$, it was determined that there was no statistically significant difference between them. This variation suggests that the AC particles possessed a greater density (2140 kg/m$^3$) than the wood particles' density. The MC of the particleboard decreased with increasing AC content, suggesting that the AC particles reduced the water uptake of the particleboards. The TS and WA of the particleboard after the 2 h and 24 h immersion tests showed different trends depending on the AC content. The particleboard with 4.5% AC had the lowest TS and WA values among all the samples, indicating that the AC improved the dimensional stability and water resistance of the particleboard by filling the pores and blocking the water penetration. However, the particleboard with 7.5% AC had higher TS and WA values than the particleboard with 4.5% AC, implying that there might be an optimal AC content for minimizing the TS and WA of particleboard. The 2 h WA and 24 h WA of the particleboards decreased with the increase in AC content, indicating that the AC improved the water resistance of the boards by reducing the capillary action and water diffusion in the board matrix. As the content of AC in the particleboards increased, it served the role of a filling material that fills voids or pores within the material. This could act as a barrier, impeding the movement of water within the particleboard, and thus, functioning as a deterrent to water ingress and absorption, consequently reducing the WA. On the other hand, the AC particles formed bonds with the wood particles, creating a more water-resistant matrix and reducing the WA. The incorporation of AC had the potential to alter the overall structure and density distribution within the particleboards. These modifications influenced how the water was absorbed and distributed throughout the material, resulting in TS and WA value changes. It was stated that in wood-based composites, the hydrophobic nature of glass wool and carbon fiber resulted in a decrease in the WA values and swelling ratios for thickness [32]. On the other hand, the addition of AC to MDF resulted in swelling ratios ranging from 15.7% to 22.4% and WA values ranging from 88% to 75% after 24 h [20,21]. Akin and Boyaci (2021) found that the TS ratios and WA values of MDF produced with UF and activated carbon were found to be between 19.46% and 22.01% and 45% and 53%, respectively [33].

Color measurements play a crucial role in assessing the color properties of wood materials and monitoring any alterations in color [34]. This method is vital for preserving the material's aesthetic appeal and ensuring quality control during production. The color changes in particleboards produced with varying ratios of AC addition can be observed in Table 3 and Figure 2.

**Table 3.** Color changes in AC-reinforced particleboards.

| Codes | L | a | b | ΔL | Δa | Δb | ΔE (%) |
|---|---|---|---|---|---|---|---|
| %0.0 AC | 58.13 | 8.44 | 24.40 | - | - | - | - |
| %1.5 AC | 61.02 | 6.30 | 22.57 | 2.89 | −2.14 | −1.83 | 4.03 |
| %4.5 AC | 59.31 | 5.18 | 20.08 | 1.18 | −3.26 | −4.32 | 5.54 |
| %7.5 AC | 54.22 | 5.33 | 18.52 | −3.91 | −3.11 | −5.88 | 7.72 |

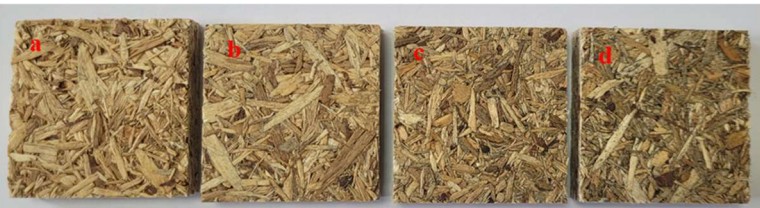

**Figure 2.** Particleboards with different ratios of AC addition: (**a**) 0% AC, (**b**) 1.5% AC, (**c**) 4.5% AC, (**d**) 7.5% AC.

Based on the data, it can be seen that the color properties of the particleboards were affected by the AC content ratio. As the AC content ratio increased, the lightness (L) and the

yellow–blue component (b) declined. It was determined that the addition of AC reduced the red–green component (a) compared to the control group (0% AC). The color difference ($\Delta E$) also increased with the AC content ratio, indicating that the particleboards became darker and less yellow as more AC was added (Figure 2d). When activated carbon was added to the composite materials, it led to a color change in the composite. Yue and Vakili (2017) found that the color of pitch-based carbon fiber composites changed from shiny gray to dark after activation with AC [35]. This result is consistent with previous studies that reported the effects of AC on the color properties of particleboards [12]. AC also has a dark color that influences the appearance of particleboards. Therefore, a trade-off between the functional and aesthetic properties of particleboards should be considered when using AC as an additive.

The microstructures of the particleboards were examined using a scanning electron microscope (SEM) at $\times 500$ magnification, as shown in Figure 3.

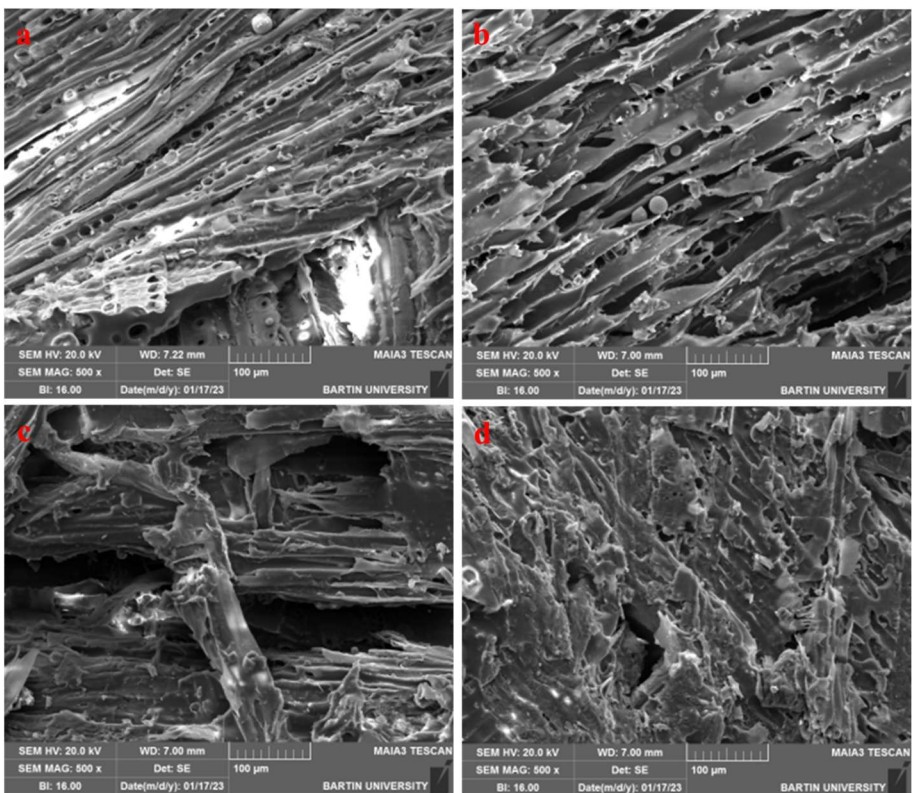

**Figure 3.** SEM images of particleboards at $\times 500$ magnification: (**a**) 0.0% AC, (**b**) 1.5% AC, (**c**) 4.5% AC, (**d**) 7.5% AC.

SEM analysis of AC-reinforced particleboards revealed that the AC was uniformly distributed within the particleboard matrix and the adhesive (Figure 3a–d). As the percentage of AC increased, the AC particles began to fill the gaps among the chips (Figure 3d). However, there was no agglomeration due to the amount of AC used, and it did not affect the mechanical performance of the particleboards. Adding fillers to the composite enhanced the mechanical properties and the surface quality of the composites by filling the voids in the polymer chains [36].

### 3.3. Mechanical Properties of Particleboards

The statistical analysis for the mechanical properties of particleboards produced with different ratios of AC addition is given in Table 4, where the letters (a and b) are used to indicate that groups of samples with similar significance are clustered together.

**Table 4.** MOR, MOE, and IB results of AC-reinforced particleboards.

| Codes | MOR (N/mm$^2$) | MOE (N/mm$^2$) | IB (N/mm$^2$) |
|---|---|---|---|
| %0 AC | 15.47 [a] ± 3.49 | 2022.06 [a] ± 404.13 | 0.70 [a] ± 0.11 |
| %1.5 AC | 16.83 [a] ± 1.86 | 2173.61 [ab] ± 249.19 | 0.73 [a] ± 0.16 |
| %4.5 AC | 17.81 [ab] ± 1.62 | 2183.58 [ab] ± 143.99 | 0.97 [b] ± 0.15 |
| %7.5 AC | 19.80 [b] ± 1.24 | 2438.92 [b] ± 131.07 | 0.99 [b] ± 0.09 |

[a,b] Letters indicate Duncan's homogeneity groups in the column.

The results show that the IB of the particleboards increased with AC content. The IB values ranged from 0.65 N/mm$^2$ for 0% AC to 0.90 N/mm$^2$ for 7.5% AC. AC improved the bonding between the particles and the resin and enhanced the mechanical properties of the particleboards. A comparison with previous studies revealed that the IB values of the particleboards with AC were higher than those of some conventional particleboards made from wood or other waste materials. The IB values of particleboards made from wood particles and UF-sodium carboxymethyl cellulose ranged from 0.28 to 0.63 N/mm$^2$, depending on the sodium carboxymethyl cellulose concentration [37]. The IB values of particleboards made from bamboo chips with UF ranged from 0.26 to 0.28 N/mm$^2$ [38]. Istek et al. (2023) found that the IB values of particleboards produced from wood particles and UF ranged from 0.59 to 0.82 N/mm$^2$ [39]. The IB of rice straw-based particleboard was determined to be 0.16 N/mm$^2$ with the addition of 4% AC [40]. AC improved the IB value of particleboards by enhancing the bonding between the particles and the resin. AC provided additional sites for chemical interactions between the resin and the carbon atoms, such as dispersion forces, electron transfer, and surface functional groups [41]. AC also increased the contact area between the particles and the resin by filling the gaps and voids in the particleboard structure [42].

The highest values of MOR and MOE were obtained for the particleboard with 7.5% AC, which were 28% and 21% higher than those without AC, respectively. When the effect of AC addition on the mechanical properties of the particleboards was examined statistically, it was understood that there were significant differences in the MOR, MOE, and IB values ($p < 0.05$). AC improved the strength and stiffness of the particleboard by reinforcing the composite material. The increase in the MOR and MOE values due to the addition of AC may be attributed to the covalent bonding between the AC and the UF resin [12]. On the other hand, oxygen-rich groups present on the surface of AC enhance mechanical strength by forming hydrogen bonds with wood fibers and adhesives [20]. Previous studies have reported different effects of AC on the mechanical properties of particleboards or other biocomposites. Maciá et al. (2016) found that AC from *Posidonia oceanica* fibers increased the MOR and MOE of biocomposites by 20% and 25% [43], respectively. However, Kibet et al. (2022) found that AC from leather waste decreased the MOR and MOE of particleboard by 10% and 12% [44], respectively. MDF with 1% AC added resulted in approximate 25% and 30% increases in the MOR and MOE values, respectively [21]. On the other hand, Basta et al. (2017) found the opposite trend for the MOR and MOE values [40] compared to our results. Some factors, such as the raw material type, chip size and geometry, amount and type of adhesive, mat moisture, board density, and the pressing conditions used in particleboard production, affect the physical and mechanical properties of wood-based boards [45,46].

### 3.4. Thermal Properties of Particleboards

The low thermal conductivity of particleboards is one of the important features desired in indoor applications. The KEM QTM 500 was utilized to obtain the thermal conductivity of particleboards using the transient hot wire method. The thermal conductivity of AC-reinforced particleboards is shown in Figure 4.

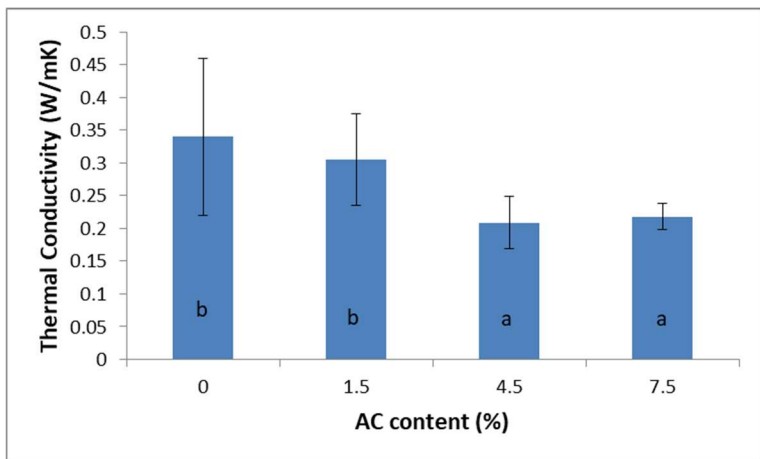

**Figure 4.** The thermal conductivity values of AC-reinforced particleboards. The letters (a and b) in Figure 4 indicate that the sample groups with equal importance are clustered together.

The results show that the thermal conductivity of the particleboard decreased with increasing AC content. The lowest thermal conductivity value (0.2155) was obtained for the particleboard with 4.5% AC, which was about 38% lower than without AC (0.3479). These results show that AC improved the insulation performance of the particleboards by reducing the heat transfer through the material. The thermal conductivity of the particleboard with 7.5% AC (0.2177) was higher than the particleboard with 4.5% AC. The thermal conductivity of a material is directly proportional to its density [47]. Previous studies have reported similar trends for other particulate composites with different types of fillers, such as graphite [48], aluminum nitride [49], and aluminum [50]. However, the thermal conductivity values of particleboards with AC in this study are lower than those reported for other composites with similar filler contents [49,50]. AC is a good thermal isolator with macro- and micro-porous structures [51]. The thermal conductivities of AC varied between 0.05 W/mK and 0.10 W/mK [52]. The thermal conductivities of wood-based panels are influenced by various factors, such as the bulk density of the chips, chip size and orientation, raw material wood species, panel density, moisture content, number of layers, and porosity [53]. Furthermore, factors such as the testing method and ambient temperature also impact the obtained values when determining thermal conductivity [54]. Hence, it was expected that commercially produced layered particleboards would exhibit different thermal conductivity values compared to single-layer particleboards manufactured in a laboratory, resulting in a more homogenous structure. In this study, the effect of AC was investigated by comparing it with particleboard without AC.

The thermal characteristics of particleboards were investigated to compare their properties with and without the inclusion of activated carbon. The analysis used the TG (Figure 5a) and DTG (Figure 5b) techniques under a nitrogen atmosphere. The findings are given in Table 5. $T_{10\%}$ indicates the temperature at which the initial 10% mass loss occurred, whereas $T_{50\%}$ represents the temperature at which 50% mass loss happened in the sample.

**Table 5.** $T_{10\%}$ and $T_{50\%}$ of thermal degradation and the mass loss (%) after thermogravimetric analysis of AC-reinforced particleboards.

| Codes | $T_{10\%}$ (°C) | $T_{50\%}$ (°C) | Mass Loss (%) |
|---|---|---|---|
| %0.0 AC | 251 | 336 | 84.51 |
| %1.5 AC | 253 | 332 | 81.45 |
| %4.5 AC | 253 | 332 | 80.11 |
| %7.5 AC | 255 | 330 | 78.36 |

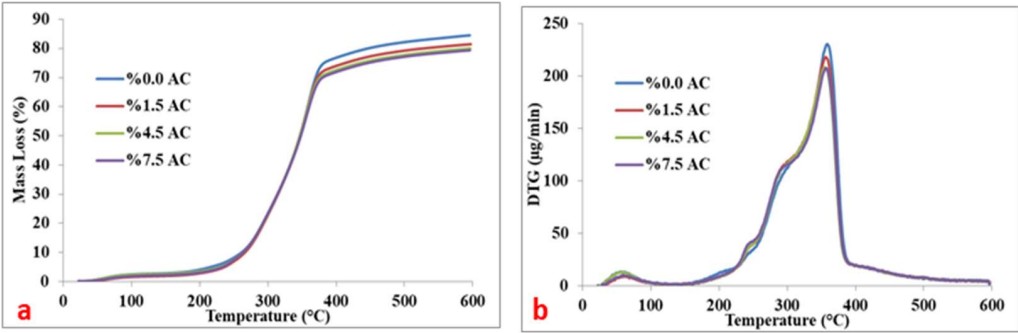

**Figure 5.** The TG (**a**) and DTG (**b**) results of AC-reinforced particleboards.

The analysis showed that $T_{10\%}$ was present between the temperatures of 251 °C and 255 °C, and $T_{50\%}$ ranged from 336 °C to 330 °C. At a thermal degradation temperature of 600 °C, the particleboard without AC (0.0% concentration) exhibited the highest mass loss of 84.51%. In contrast, the particleboard with a 7.5% AC concentration had the lowest mass loss of 78.36%. These results indicate that the addition of AC had a positive effect on the thermal stability of the particleboard. AC, consisting of graphitized carbon layers, could act as a physical barrier, limiting the diffusion of heat and oxygen during the combustion process [55].

TGA and DTG analyzed the compositional and thermal properties of different materials, such as particleboards made from wood particles bonded together with adhesives [56]. Particleboards had different thermal degradation kinetics depending on the type and amount of adhesive used, the particle size and distribution, the pressing conditions, and the presence of additives or fillers [57]. The thermal decomposition of particleboards involved three main stages: moisture evaporation, degradation of hemicellulose and cellulose, and degradation of lignin and resin. The first stage occurred below 100 °C, the second stage occurred between 200 °C and 400 °C, and the third stage occurred above 400 °C [58]. The TGA/DTG curves of the particleboards showed different peak positions, shapes, and intensities depending on the composition and structure of the boards. Particleboards manufactured from natural rubber latex-based bio-adhesive showed a single DTG peak at around 300 °C, while particleboards made from phenol-formaldehyde resin showed two DTG peaks at about 345 °C and 435 °C [59]. As the amount of AC added to the particleboards increased, the burning rate of the material decreased, which can be observed in the peaks in the DTG analysis (Figure 5b).

In our study, the utilization of activated carbon demonstrated improvements, particularly in the mechanical and thermal properties. With its high surface area, rich functional groups, and porous structure, activated carbon's incorporation into the production of wood-based panels holds significance in enhancing the in-service performance of these panel products. Moreover, the potential utilization of low-value or waste lignocellulosic materials in the panel industry, facilitated by AC, adds further importance. This approach has implications for supplementing the raw material base for the panel industry.

## 4. Conclusions

The incorporation of AC in particleboards demonstrated various effects on their properties. Our analysis revealed that an optimal AC concentration of 4.5% led to the best result of enhanced physical properties, including water resistance and dimensional stability. The addition of AC caused a total color change ($\Delta$E) of up to 7.72%, so care must be taken when applying AC in situations where aesthetics are important. There were no statistical differences between the 4.5% AC addition and the 7.5% AC addition in the IB, MOR, and MOE properties. Compared to the reference particleboards, the addition of 7.5% AC improved the IB, MOR, and MOE values of the particleboards by approximately 41%, 28%, and 21%, respectively. The thermal properties of the particleboard enhanced with increasing AC content.

All results show that an optimal AC concentration existed for achieving a balance among improved physical, mechanical, and thermal properties while considering aesthetic changes. The particleboard containing 4.5% AC consistently demonstrated favorable outcomes across multiple properties. The addition of 4.5% AC could be considered an optimal concentration to achieve a synergistic enhancement in the overall performance of particleboards for indoor applications.

**Author Contributions:** M.E.E.: conceptualization, investigation methodology, writing—review and editing; İ.Ö.: investigation methodology, writing; A.İ.: editing, supervision; A.C.: methodology, writing—review and editing. All authors have read and agreed to the published version of the manuscript.

**Funding:** This research received no external funding.

**Institutional Review Board Statement:** Not applicable.

**Informed Consent Statement:** Not applicable.

**Data Availability Statement:** The data presented in this study are available upon request from the corresponding author. The data are not publicly available due to further research activities.

**Conflicts of Interest:** The authors declare no conflict of interest.

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
