# Peer review of "Analysis and Impact of Activated Carbon Incorporation into Urea-Formaldehyde Adhesive on the Properties of Particleboard"

_coatings, doi:10.3390/coatings13091476_

Round 1
Reviewer 1 Report
In this manuscript, AC as filler was added with different dosage in the UF resin to prepare the particle board. The effect of AC on the physical and mechanical properties of PB was detected and compared. Some necessary experiments and analyzes have been carried out. However, there are still problems with experimental methods and data processing.
Some specific comments are shown as below.
1. What is the particle size of activated carbon used?
2. L86: “ The adhesive-coated chips were subsequently sprayed onto the chips within a rotary drum mixer, ensuring…” Is the statement correct about the method of sizing?
3. Supplement the number of samples for each test analysis, and statistical analysis of data should be done.
4. Supplement color analysis and inspection equipment description. Is the data in Table 3 calculated correctly? And the data analysis in the paper is inconsistent with the data in the table 3.
5. How to explain the decrease of thickness expansion and the increase of water absorption with the increase of activated carbon addition? The results of the literature can not replace the discussion for the phenomenon that the paper needs to analyze and explain.
6. The data in Figure 2 and Table 4 are duplicated.
7. L268: “highest mass loss of 84.51%, while the foam with a 7.5% activated carbon concentration” What does the foam mean?
8. Particleboard generally needs a secondary finish as a substrate, so it is more meaningful to study the wettability of the board surface than the color.
Author Response
Dear Reviewer,
We would like to thank you for the insightful comments and suggestions. We made all possible changes that were suggested and detailed the changes in the table below. Prior to response your comments we want to inform you that all the revisions and improvements are highlighted yellow color in the revised version of our manuscript. We sincerely appreciate your insightful comments on our paper. We would like to thank you again for your valuable time and insight to strengthen our paper.

Reviewer 2 Report
Investigating the Impact of Activated Carbon on Particle Board Properties: A Comprehensive Analysis
Coatings-2550300
Dear Editor,
I have read the Manuscript and referred to some suggestions and comments. Thanks to the authors for doing a good study. The work will be more suitable for readers if the comments included.
Title: No need to put words “Investigating the…”
Abstract: It should be better to write a sentence about the AC at the beginning of the abstract. What the Ac is?
L. 16: Plz report that how the physical properties and the other features measured. Name the instruments.
Introduction
l. 37 up to 43 just on reference. It cannot be suitable.
l. 64: the aims of the study have been stated in a fragile manner. I suggest the authors empower the last para of the Intro.
2. Materials and Methods
L. 72: Could you define the chips portion of broad-leaved trees? The approximate percentage of each?
Where did the authors produce the PB? Was the PB made in the Lab. Or based on the company level and features?
L. 85: How was the mixture stirred? Name the instrument model.
L. 87: write the name and the model of the rotary drum.
L. 94: Why “Can be given’? It’s better to rewrite it as “is shown”.
L. 99: You should refer to the Standards in the Reference Section. Every standard is a reference.
L. 107: The SEM samples were dried or scanned with their usual moisture? If they dried, plz define how.
L. 114: What are TG and DTG? Put their full format first.
Did the researchers run any statistical analyses? If yes, must put the details at the end of Section 2.
3. Results and Discussion
All results must be supported by statistical analysis.
The items in Table 2 must be superscripted by significant letters.
L. 122-123: must be determined by the level of probability.
L 125: once the abbreviations are used, the rest just put the abbreviations. This must be run for all abbreviations.
In the Results and Discussions Section must put the importance of AC. Why did the AC have value for researchers?
The tense of all verbs in Results must be in past because the work finished.
L. 142-143 must be supported by a Reference.
Table 3: put the significant letters.
Describe the abbreviations used in the Table in the caption. ΔL?
L. 163: What results? The new paragraph is a new idea.
Table 4: same as the other Tables must be statistically determined.
Figure 2: Put the AC in the X-axis for all valuables.
L. 207: Every result must support by a reference. You should define why these types of results happened. Must bring the reason (s) for each result.
L. 211: All scientific names must be in Italic format.
Figs 3 and 4: Why is there a difference between the types of columns?
What are the numbers on the columns?
L. 233: must be supported by a reference.
L. 238: must be supported by a reference.
L. 240 same as above.
The authors should define the AC features and characteristics. If there is a SEM image for Ac, it suggests inserting it.
L. 247: support the report by refereeing to the related Figures/Tables.
L. 255: What is the HDPE? Full format.
The statistical analysis must support table 5.
Ti and Tmax?
I think, there are no differences between the values determined by the authors. The data must be statistically analyzed.
L. 269: The results showed positive impacts of AC on the PB, but I doubt it wouldn’t be significant.
All Figs/Tables must be cited in the text. Plz do it for the entire text.
L. 292: refer the result to the specified Fig/Table.
Conclusion
Shorten this section. Put the most important findings.
Dear Editor, the quality of the English language is good but needs some minor corrections.
Author Response

(The authors gave the same response as above.)

Reviewer 3 Report
Dear authors,
I have read your manuscript. My detailed comments are highlighted in the attached file. Please note two major points in particular. 1. The reason for adding AC to the PB is not clear in your text. 2. It is known that AC acts as a formaldehyde adsorbent for wood-based panels. However, this was not measured and reported in your manuscript.
Best wishes

Author Response
Dear Reviewer,
We would like to thank you for the insightful comments and suggestions. We made all possible changes that were suggested and detailed the changes in the table below. Prior to response your comments we want to inform you that all the revisions and improvements are highlighted yellow color in the revised version of our manuscript. We sincerely appreciate your insightful comments on our paper. We would like to thank you again for your valuable time and insight to strengthen our paper.
The reason for adding AC to the PB is not clear in your text. Necessary explanations were made in the text
It is known that AC acts as a formaldehyde adsorbent for wood-based panels. However, this was not measured and reported in your manuscript. The effects of adding activated carbon to MDF were investigated and have limited studies in relation to formaldehyde emissions. However, our current study is focused on examining the physical, mechanical, and thermal properties on particle board, with ongoing research on formaldehyde emissions. We intend to present the investigation of formaldehyde emissions as a distinct and long-term study to explore its prolonged impact.
My detailed comments are highlighted in the attached file. Necessary correction and explanations were made in the text.

Reviewer 4 Report
The topic is actual and interesting. The manuscript is well structured and discussed, however it have some disadvantages indicated below. After the amendments the manuscript should be peer-reviewed and considered for publication in Coatings.
Title: Since the manuscript excludes the important analysis of formaldehyde emission, I suggest to rename it as following: "Analysis and impact of activated carbon incorporation to urea-formaldehyde adhesive on the properties of particleboard".
Abstract: the abstract should include the used AC rate and the optimal/suggested concentration.
Introduction: Introduction is good enough, however, since AC is in investigation stage for particleboard and fiberboard for at least a decade, there should be given a clarification about what have been done and what new your paper contributes to the investigation in the field.
Line 39: please use one term of "particleboard" throughout the manuscript instead of divided "particle board(s)"
Line 43-44: since the reference allow you can mention here the amount of produced particleboards not only in Turkey, but in the world as well.
Line 45-51: please clarify your statement “are used” - do you mean these mentioned fillers are used in the industry or in the investigation stage?
Section 2.1: is it is possible please indicate whether origin of AC was organic or inorganic.
Section 2.2: please indicate in the subsection how many board samples were produced per each sample group? Since the used wood chips contained softwood and even several hardwoods, please indicate the used ratio of each species and its dimensions.
Section 2.3: please include in the Section the color measurement details as well.
Table 1: Please clarify in the text the variation of pressure and pressing time.
Line 99-106: since the national standards are derived from European Norms (EN) or International Standardization Organization (ISO) it is enough to mention only EN 326-1 (1999) or ISO 554 (1997).
Line 104: please add "...and thickness swelling (TS)..."
Line 111-113: please indicate the temperature range within which the thermal conductivity was measured.
Results and Discussion: it is suggested to split the Section to: 3.1 Physical properties, 3.2 Mechanical properties (IB, MOE, MOR) and 3.3 Thermal properties (thermal conductivity, and TGA).
Table 2: Please supplement the results by standard deviation and correct the density values (750 instead of 0.75) according to the unit in kg/m3.
Line 122-136: from the values of Table 2 (I guess there are average values) it could not be said that the results are influenced by AC. Only in the case of TS after 2h with 1.5% of AC the difference is higher with reference sample. However, to evaluate the values scientifically to state whether the differences are or not significant, at least the analysis of variance (ANOVA) should be performed!
Line 136-141: The discussion could be supplemented also by https://www.researchgate.net/publication/357771832_Effects_of_Activated_Carbon_on_Medium_Density_Fiber_Board_Properties.
Line 151-153: Please correct the statement because the lightness first increases and only slightly decreases at AC 7.5%. In turn, component (a) shows a linear decrease, not increase as you state.
Line 155: based on your and other authors observation write better following: "it causes the color change of the composite".
Line 168: please correct the text because the density is not given in Table 4, but in Tble 3.
Table 4: please add the standard deviation of the results. Otherwise, there is no evidence on the improvement by the addition of AC.
Line 187: please use one unit for mechanical properties - N/mm2.
Line 190: the efect of AC on IB is not so clear because, for e.g., Akin and Karaboyachi 2021 (https://www.researchgate.net/publication/357771832_Effects_of_Activated_Carbon_on_Medium_Density_Fiber_Board_Properties) declare that the increase of AC from 1% to 5% results to the decrease of IB values from 0.34 to 0.3 N/mm2.
Figure 2: The figure is not necessary, because the presented results repeat the results summarized in Table 4.
Line 209-219: here you can mention your cited Basta et al. 2017 which declared the opposite trend on MOR /MOE than yours. This means that the effect of AC addition is highly-dependent on a lot of factors like AC concentration, adhesive formulation, the method of AC incorporation, wood particles origin and their dimensions, and finally the pressing cycle. So, the discussion could be supplemented by these considerations based on the literature sources.
Figure 3: How would you explain the obtained high thermal conductivity values taking into account that for commercial particleboard these are 0.12-0.18 W/mK? (https://www.researchgate.net/publication/263300577_Agricultural_and_Industrial_Valorization_of_Arundo_donax_L). Please add the standard deviation in the figure and make a statistical analysis to evaluate the significance of differences between the samples.
Line 250-251: this statement should be highlighted in the Figure 4. The current presentation of the figure does not show the presence of AC. Therefore, it is suggested to increase the dimensions of each figure to see the differences these could be supplemented by arrows indicating the obvious presence of AC, particularly indicating the filling of gaps between the chips.
Table 5: please define Ti and Tmax before in the text.
Line 268: “foam” - do you mean "the board sample"?
Line 290-292: If the statement is related to the Figure 5, please give the reference to it.
Conclusions: Please rewrite the Section after a statistical analysis for each performed property results. Based on the results given in Table 2, your conclusion made for the red-green component is wrong. The conclusions should include the optimal AC concentration that is on your decision. It is suggested to divide the section to the separate paragraphs.
Though English is not my mother tongue, the quality of English is good enough and understandable. Only minor editing of English language required.
Author Response

(The authors gave the same response as above.)

Round 2
Reviewer 1 Report
The manuscript has been revised according to the review comments, however, there are still some mistakes in the manuscript. The suggestions are as follows:
1. Abstract: The first two sentences in the abstract are suggested to be deleted because AC is an outsourced material, there is no need to describe its preparation method and characteristics in the abstract.
2. L91:“The dimensions of the chips were 10.5mm x 10.5mm.” Is the size of wood chip right?
3. L196-197: "This variation suggested that AC particles possessed a greater density (214 kg/m3) than wood particles' density." What is the density of wood particle? Is the causal relationship correct?
4. Why is the TS value of particleboard added with 7.5%AC greater than that of particleboard added with 4.5% AC?
5. L360: Figure 6 should be Figure 5.
Author Response
Dear Reviewers,
We would like to thank you for the insightful comments and suggestions. We made all possible changes that were suggested and detailed the changes in the table below. Prior to response your comments we want to inform you that all the revisions and improvements are highlighted yellow color in the revised version of our manuscript. We sincerely appreciate your insightful comments on our paper. We would like to thank you again for your valuable time and insight to strengthen our paper.

Reviewer 2 Report
Thank you for the authors to do their best in responding to the comments. The Paper is now good.
Author Response
We would like to thank you for the insightful comments and suggestions. Thank you
Reviewer 3 Report
Dear respected Authors,
Thank you for your revision. But I disagree with one of your statements.
Page 3, L 106-108: You mentioned that the viscosity did not change after adding AC. How is it possible that the viscosity did not change after adding 7.5% of a voluminous material like AC to the resin? The solids content of your adhesive was 65% before the addition of AC. By adding 7.5% AC, you definitely have a more viscous binder which makes it more challenging for resin spraying.
Best regards
Author Response

(The authors gave the same response as above.)

Reviewer 4 Report
Dear authors,
Thanks for providing improvements. The manuscript is good enough, however, some amendments are still suggested. These are indicated in the attache file.

The English language of the manuscript is good enough and could be easy to understand. Only in some places indicated in the attached file the amendments should be done.
Author Response
Thanks for providing improvements. The manuscript is good enough, however, some amendments are still suggested. These are indicated in the attached file.
Necessary corrections were made in the pdf file.
